# Secure and Privacy-Preserving Authentication Scheme Using Decentralized Identifier in Metaverse Environment

Myeonghyun Kim [1], Jihyeon Oh [1], Seunghwan Son [1], Yohan Park [2], Jungjoon Kim [3] and Youngho Park [1,*]

1　School of Electronic and Electrical Engineering, Kyungpook National University, Daegu 41566, Republic of Korea; kimmyeong123@knu.ac.kr (M.K.); chldlstnr071@knu.ac.kr (J.O.); sonshawn@knu.ac.kr (S.S.)
2　School of Computer Engineering, Keimyung University, Daegu 42601, Republic of Korea; yhpark@kmu.ac.kr
3　School of Electronics Engineering, Kyungpook National University, Daegu 41566, Republic of Korea; jungkim7@ee.knu.ac.kr
*　Correspondence: parkyh@knu.ac.kr

**Abstract:** The metaverse provides a virtual world with many social activities that parallel the real world. As the metaverse attracts more attention, the importance of security and privacy preservation is increasing significantly. In the metaverse, users have the capability to create various avatars, which can be exploited to deceive or threaten others, leading to internal security issues. Additionally, users attempting to access the metaverse are susceptible to various external security threats since they communicate with service providers through public channels. To address these challenges, we propose an authentication scheme using blockchain, a decentralized identifier, and a verifiable credential to enable metaverse users to perform secure identity verification and authentication without disclosing sensitive information to service providers. Furthermore, the proposed approach mitigates privacy concerns associated with the management of personal information by enabling users to prove the necessary identity information independently without relying on service providers. We demonstrate that the proposed scheme is resistant to malicious security attacks and provides privacy preservation by performing security analyses, such as AVISPA simulation, BAN logic, and the real-or-random (ROR) model. We also show that the performance of our proposed scheme is better suited for the metaverse environment by providing greater security and efficiency when compared to competing schemes.

**Keywords:** metaverse; authentication; blockchain; decentralized identifier

## 1. Introduction

Various advanced technologies are rapidly evolving and being invented, leading to the emergence of the metaverse concept, which is envisioned as the next iteration of the Internet. Metaverse is a virtual realm that parallels the physical world, where people engage with the metaverse using wearable devices (such as a virtual reality (VR)/augmented reality (AR) devices) and manipulate digital avatars to engage with others. Furthermore, the advancement of cutting-edge communication and networking technologies, including wireless networks and 5G technology, plays an important role in moving the metaverse forward by enabling low-latency, high-speed, and reliable data exchange between devices and the network. In addition, AI technology also contributes to automating the creation of virtual environments and digital items, and extracting valuable insights from the vast amount of data generated within the metaverse [1,2]. Blockchain, serving as a trust infrastructure in decentralized distributed networks, enables individual-centric digital asset transactions for metaverse users, not tied to traditional service providers' platforms. It can also contribute to achieving the compatibility of individual services held by various virtual spaces (or service providers) within the metaverse [3]. The metaverse is anticipated

to bring about great innovation in various aspects of life, including e-commerce, medical, education, entertainment, smart factory and other social services [4,5].

In the metaverse, users can create avatars to represent themselves virtually, and they can access various services through these avatars. However, in the current metaverse application, users possess the freedom to create any avatar to serve as their virtual representation, irrespective of their real-world identity. This characteristic presents avenues for malicious users to fabricate a similar avatar and cause serious security problems, such as identity leakage, theft, and virtual asset fraud during avatar interactions. In addition, issues such as stalking, harassment, and sexual assault can pose a threat to users by manipulating the avatar, as well as the potential privacy threat of using AI technology to monitor users, make inferences about them, or engage in impersonation [6–8]. Furthermore, users need to exchange their information and data with third parties to access services offered in various virtual worlds within the metaverse. However, due to the aforementioned characteristics, the identity information of the third parties using the user's information is often unclear, making interactions for users challenging. Examples include qualifications to provide professional services such as medical or educational services, or adult verification to use certain data. Therefore, it is essential to design an authentication scheme that allow users to safely use services in the metaverse and remain secure from other security threats.

In current metaverse application, users have no direct means to verify the identity of other avatars as malicious or not, so they need help from the metaverse service provider. In the process of tracking these manipulators, the service provider mainly utilizes the manipulator's account and password as clues to track the manipulator from a specific avatar identity [9]. However, employing password-dependent methods means that any player who knows the account password can successfully gain access, so if a malicious user obtains the password illegally through various means, he/she can log in illegally and manipulate the avatar of a legitimate player. For more secure user identification and assurance on the metaverse, users can provide a lot of personal information to service providers. However, service providers that collect sensitive information, such as users' voices and motions generated in the metaverse, can abuse this personal information and cause users' privacy violations and huge losses through advertisements, personal tracking, fraud, illegal use, etc. In addition, the users and platform servers communicate through public channels in metaverse environments. Thus, an external adversary can attempt to eavesdrop and forge messages transmitted over public channels and attempt various security attacks, including masquerade, replay and man-in-the-middle attacks. Therefore, sensitive user information should not be disclosed to external parties and should only be shared with specific stakeholders in specific circumstances.

In this paper, we propose a blockchain-based authentication scheme that utilizes decentralized identifiers and verifiable credentials technology to enhance system security and protect users from various security and privacy threats. Decentralized identifiers and verifiable credentials enable trustworthy identity verification and data exchange without intermediaries. We propose an authentication scheme where users can authenticate not only avatars but also real manipulators during the authentication process required before interactions between avatars, using the users' decentralized identifiers and verifiable credentials. Additionally, to ensure secure communication and avatar interactions in the metaverse environment, we propose an authentication method using blockchain between users and platform servers and between avatars. In our proposed scheme, the user and service provider establish security communication channels during the login phase through secure authentication and key agreement. Furthermore, we minimize user information exposed to service providers during interactions with other avatars and enhance user privacy protection by allowing only the necessary personal identification information for verification when interacting with different avatars in the metaverse.

Furthermore, in the metaverse, during the consensus process of validating and recording information on the blockchain, security attacks, such as 51% attacks and Sybil attacks, can occur [10–12]. These attacks can undermine the trustworthiness of information recorded

on the actual blockchain. However, in this paper, the consensus process occurs only once when the user initially creates a unique ID and registers it in the system. Subsequently, during the authentication process, users verify the required record information on the blockchain, and at this point, the blockchain's consensus process does not occur, minimizing the consensus process. Additionally, this paper assumes the security of the blockchain consensus process and focuses on security threats and privacy issues during the user registration phase and subsequent use of metaverse services.

### 1.1. Contributions

The main contributions of paper are as follows:

- In the metaverse environment, users are exposed to threats, such as fraud through fake avatars and the risk of personal information leakage during data transmission through open channels. We propose a secure authentication method for the metaverse environment to ensure security against various threats arising from fake avatars or vulnerabilities in wireless communication channels, and provide forward secrecy, anonymity, and privacy preservation.
- The proposed scheme utilizes decentralized identifiers and verifiable credentials to enhance user privacy protection. Metaverse users can provide only the necessary identity information to stakeholders without disclosing their information to external parties, thereby safeguarding their personal information.
- We perform an informal analysis to ensure that the proposed scheme can provide security against various attacks, including impersonation, session key disclosure, replay, man-in-the-middle, and insider attacks. Additionally, we show that the proposed scheme can achieve mutual authentication, perfect forward secrecy, anonymity and privacy preservation.
- The security of the proposed scheme is analyzed by performing informal and formal analyses, such as Burrows–Abadi–Nikoogadam (BAN) logic, the real-or-random (RoR) model, and the automated validation of internet security protocols and applications (AVISPA) simulation tool. We also compare the performance and security features with the related works to show that the proposed scheme is superior.

### 1.2. Organization

The organization of the paper is as follows. Section 2 reviews the existing authentication scheme applicable to the metaverse environment. Section 3 introduces relevant preliminaries. Section 4 presents a proposed system model and adversary model. The details of the proposed authentication scheme are depicted in Section 5. Section 6 analyzes the security of the proposed scheme in informal and formal proofs, and Section 7 analyzes the computation and communication costs of the proposed scheme and related works. Finally, we summarize the conclusion and the future works in Section 8.

## 2. Related Work

With the emergence of metaverse platforms (e.g., roblox and minecraft) and the increasing number of applications that utilize the metaverse, the security of the metaverse environment is discussed in several studies [13–15]. According to the paper proposed by Vu et al. [13], in the virtual world, users may find themselves in a situation where they are required to present identity information in order to obtain certain services and activities. They argued that not only are authentication mechanisms required to ensure that metaverse users can access the platform with appropriate identities but IoT devices in the metaverse infrastructure (e.g., sensors and UAVs) also need effective mechanisms for authentication during operation. They asserted that blockchain technology can address metaverse security and privacy issues, including identity and authentication management. Patwe and Mane [14] argued the necessity of designing a secure authentication mechanism because impersonation, server spoofing, mutual authentication threats, and replay attacks can occur in the metaverse environment. And they proposed a blockchain-based architec-

ture for avatar and user authentication in consideration of the decentralized nature of the metaverse. However, to date, there are no proposed specific system models and mutual authentication schemes for metaverse environments.

In the metaverse environment, where users use virtual services from the service provider's server using wearable devices, such as VR and AR, some mutual authentication methods for the IoT environment can be applied. Panda and Chattopadhyay [16] proposed an elliptic curve cryptography-based mutual authentication protocol to ensure secure communication between IoT devices and cloud servers. They argue that the proposed scheme is secure against various security threats (including impersonation attack, replay attack, etc.) by performing an informal analysis and using the AVISPA simulation tool. However, they did not consider the device-hijacking attack scenario. In the metaverse, there is a risk of maliciously capturing and tampering with a user's XR device to extract sensitive information or impersonate a legitimate user to gain access to the system. Li et al. [17] proposed a mutual authentication scheme based on blockchain for users and servers. Li et al.'s scheme solves the problem of SPoF that occurs in the centralized authentication structure by proposing a blockchain-based decentralized authentication scheme. They claimed that their scheme is secure against impersonation and man-in-the-middle attacks, and that it also provides perfect forward secrecy. However, security features such as insider attacks and anonymity are not covered. These schemes can be applied to authentication between a user's device and a service provider's server. However, it is difficult to apply these schemes to the authentication mechanism required for interactions between avatars in the metaverse environment. Ryu et al. [18] proposed an authentication scheme that can ensure secure communication in a metaverse environment and transparently manage user identification data using blockchain technology. They designed the necessary mutual authentication methods to provide secure communication between platform servers and users as well as secure interactions between avatars. However, users who manipulate avatars in the metaverse need to prove their real-world information (e.g., age, gender, occupation and account) to other avatars in specific situations. Ryu et al.'s avatar authentication scheme can expose a lot of personal information of users to metaverse service providers. If personal information is exposed, it is possible to track the avatar's user, or to impersonate a legitimate user by using a camouflage avatar.

Therefore, there is a need for research on authentication methods that can provide secure communication and privacy protection for users while considering the characteristics of the metaverse. We propose an authentication and key agreement scheme to enable metaverse users to securely utilize services from service providers. Furthermore, within the platform, we propose a secure authentication scheme between avatars that allows users to protect their privacy during avatar interactions without relying on the service provider.

## 3. Preliminaries

This section briefly introduces a fuzzy extractor, decentralized identifier (DID) and verifiable credential (VC).

### 3.1. Fuzzy Extractor

The fuzzy extractor [19] is widely acknowledged for confirming biometric validation. A biometric key can be constructed using a biometric outline, such as irises, facial features, and fingerprints. The characteristics of the fuzzy extractor are defined by the following two algorithms, including a probabilistic algorithm $Gen(\cdot)$, and a deterministic algorithm $Rep(\cdot)$:

- $Gen(BIO) = (r, \delta)$: The user's biometric information $BIO$ is accepted as an input parameter to the algorithm. Then, the secret value $r$ is output along with the public reproduction parameter $\delta$.
- $Rep(BIO, \delta) = (r)$: The algorithm accepts a noisy user biometric $BIO$ from the user, controlling the noise using the public reproduction parameter $\delta$. Then, $Rep$ reproduces the original biometric secret value $r$.

*3.2. Decentralized Identifier and Verifiable Credential*

The decentralized identifier [20] is a concept designed to uniquely identify the digital identities of users and entities within a distributed network. It allows users to manage and verify their identities in a decentralized manner, without relying on central identity verification authorities. Users can confirm or show their DID ownership by employing cryptographic methods, such as digital signatures. DIDs are stored in conjunction with blockchains, ensuring their immutability and security. The features and operation of DIDs in the proposed scheme are as follows:

1.  **Decentralized identifier creation**: Users or entities generate DIDs. DIDs are unique and can be created by users themselves, not centralized authentication authorities.
2.  **Integration with blockchain**: DIDs are stored in conjunction with a blockchain. This ensures that DIDs are stored in a distributed registry, making duplication or alteration difficult.
3.  **Digital identity verification**: To log in to digital services or applications using their DID, users create a signature using their private key.
4.  **Distributed identity management**: Users manage their DIDs and identity information in a distributed network. This information is stored on the blockchain, ensuring immutability, and users share it only when necessary.

A verifiable credential [21] is a concept and technology used to represent and verify personal identities and permissions in the digital realm. Verifiable credentials serve as an alternative to centralized identity verification systems, allowing individuals to manage and share identity information (credentials) issued by identity authorities. The features and operation of VCs in the proposed scheme are as follows:

1.  **Creation of VCs**: Users process their identity-related data to generate VCs. These VCs include the user's identity information and the user's signature using the elliptic curve-based signature algorithm.
2.  **Issuer of VCs**: VCs are created by the party or institution that issues the information. The issuer verifies the source of the information and signs the VC to ensure its integrity.
3.  **Storage and transmission of VCs**: VCs are stored in a digital format, and users share them only when necessary. VCs are securely transmitted and stored, often in encrypted form.
4.  **Verification of VCs**: When presenting VCs to a verifier, the verifier uses the issuer's public key to verify the signature of the VC and validate the accuracy of the information. This confirms the authenticity of the VC.
5.  **Selective sharing of VCs**: Users can share only the necessary information through VCs, enhancing personal data protection. They provide minimal information to third parties and perform required identity verification.

## 4. System Model

Our proposed secure and privacy-preserving authentication scheme using a decentralized identifier in the metaverse environment is composed of four entities, including certificate authority, service provider, user, and blockchain. We depict the proposed system model in Figure 1, and describe each entity as give below.

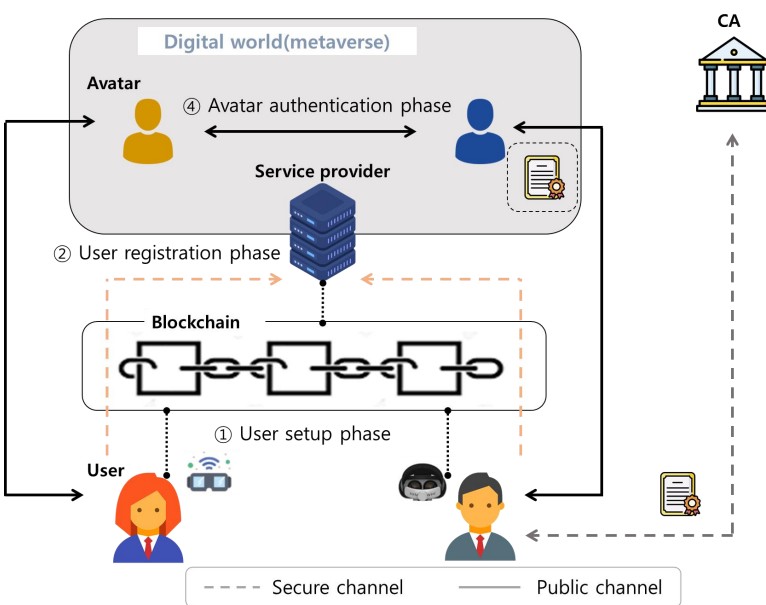

**Figure 1.** The proposed system model.

- Certificate authority ($CA$): $CA$ serves as a fully trusted entity that initializes and publishes system parameters. $CA$ receives the user's decentralized identifier and personal information, which require verification. Then, $CA$ verifies both and issues a credential to the user proving the user's personal information (occupation, age, etc.). The credential values must be authenticated between the users/avatars in the metaverse environment.

- Service provider ($SP$): $SP$s offer services that enable users to engage in various activities in virtual spaces, such as education, gaming, healthcare, and more. The user first registers on the $SP$ using the decentralized identifier. If a user attempts to access the $SP$, $SP$ verifies the correct identity of the user. In addition, the $SP$ is responsible for forwarding request and response messages that occur in its own virtual space during the avatar authentication phase.

- User: The user creates his/her own decentralized identifier on the blockchain. The user sends his/her decentralized identifier and personal information to $CA$ to receive credentials to prove their personal information. Then, the user registers with the $SP$ to participate in the metaverse environment. At this time, the user transmits only minimal information to register with the $SP$, and no other personal information is transmitted. The user can interact with other users by using avatars created in the virtual world, such as exchanging information with other users for various purposes. The user uses DID, public key, and verifiable credentials in the virtual space to mutually authenticate with other users' avatars to achieve secure interaction between avatars and avatars.

- Blockchain: In the proposed authentication scheme, we adopt the public blockchain, which is a fully decentralized infrastructure. In the public blockchain network, every node can easily join blockchain networks without the need for a trusted authority. All blockchain members can read the ledger and upload transitions to the blockchain. To ensure that all entities participating in the system agree on a single source of truth, the public blockchain adopts proof-based consensus algorithms, including proof of work and proof of stake. In our system, the blockchain is adopted to store the information required for authentication, and it does not contain any other information other than DID documents. In the proposed scheme, we assume that the consensus process of the blockchain operates correctly and reliably.

The process flows of the proposed scheme are described as follows:

- **User setup phase:** The user generates their own decentralized identifier. The $CA$ issues a verifiable credential to the user that proves the user's personal information.

- **User registration phase:** The user registers with the *SP* using his/her own decentralized identifier. The *SP* verifies that the user's decentralized identifier is valid, and then the user's avatar is generated in virtual space.
- **Login phase:** When the user attempts to access the *SP*, the user and *SP* authenticate each other. If the mutual authentication between the user and *SP* is completed and the session key is agreed upon, the user and *SP* establish a secure communication channel through the session key.
- **Avatar authentication phase:** In the virtual space, the user can interact with other avatars. For secure avatar-to-avatar interactions, the user provides verifiable credentials, proving the personal information needed to perform the avatar authentication phase.

### 4.1. Adversary Model

The adversary can have the following capabilities based on the Dolev–Yao (DY) threat model. The Dolev–Yao threat model [22] is widely employed in the analysis of protocol security [23–25]. The capabilities of an adversary are defined as follows:

- An adversary can eavesdrop, intercept, modify, expunge, and forge the transmitted messages through a public channel.
- An adversary can conjecture about either the identity or the password of a legitimate user, but it is incapable of conjecturing about both simultaneously.
- An adversary can physically seize the user's XR devices and infer sensitive data through power analysis attacks [26–28].
- An adversary can attempt to launch various attacks, including impersonation, replay attacks, and man-in-the-middle attacks.
- An adversary can be an insider in the *SP*.

For this work, we also adopt a more stringent adversary model, known as the "Canetti–Krawczyk (CK) model" [29]. In the CK model, the adversary not only has all the capabilities of the DY model but the adversary can obtain ephemeral session states and long-term values (including secret keys) by performing a session-hijacking attack. The adversary also creates a replica avatar in the metaverse environment to deceive others.

## 5. Proposed Scheme

This section presents the proposed secure and privacy-preserving authentication scheme using a decentralized identifier for the metaverse. The proposed scheme includes the initialization, user setup, registration, login, and avatar authentication phases. Table 1 describes the symbols used in the scheme.

**Table 1.** Symbols and their meanings.

| Symbol | Description |
|---|---|
| $U_i$ | *i*-th user |
| $SP$ | The service provider |
| $CA$ | A certificate authority |
| $ID_i, PW_i$ | Identity and password of $U_i$ |
| $sk_x, PK_x$ | Secret key and public key of entity $x$ |
| $DID_x$ | Decentralized identity of entity $x$ |
| $H(\cdot)$ | Hash function |
| $T$ | Timestamp |
| $\alpha_i, \beta_x, x_x, a_x$ | Random nonces |
| $\oplus$ | XOR operation |
| $\parallel$ | Concatenation operation |

### 5.1. Initialization Phase

First, $CA$ initializes the system parameters. $CA$ generates large prime numbers $p, q$, an additive group $G$, elliptic curve $EC_p$ over $F_p$, a generator $P$, one-way hash functions $H.$, and

a secret key $sk_{CA}$, and it computes a public key $PK_{CA}$ corresponding to $sk_{CA}$. After that, $CA$ publishes the system parameters $par = \{p, q, G, EC_p, P, PK_{CA}, h(\cdot)\}$ to the network.

### 5.2. User Setup

The user generates their own decentralized identifier. $CA$ issues a verifiable credential to the user that proves the user's personal information. This phase is performed over a secure channel. Figure 2 shows the user setup phase and detailed processes steps are as follows.

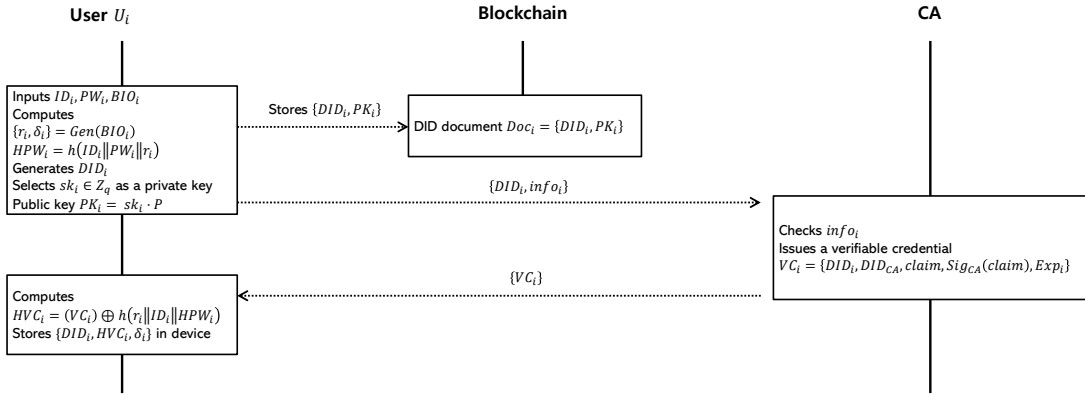

**Figure 2.** User setup phase of the proposed scheme.

- **US-1:** User $U_i$ inputs a unique $ID_j$, password $sk_j$ and biometric information $BIO_i$. Then, $U_i$ selects a random number $sk_i \in Z_q$ as a private key and computes $Gen(BIO_i) = \{r_i, \delta_i\}$, $HPW_i = h(ID_i||PW_i||r_i)$, $PK_i = sk_i \cdot P$. Then, $U_i$ generates the $U_i$'s own $DID_i$ that indicates the location of the DID document $Doc_i = \{DID_i, PK_i\}$ on the blockchain.
- **US-2:** $U_i$ requests $CA$ to issue a credential by sending $DID_i$, personal information $info_i$. $CA$ checks a $U_i$'s personal information and $DID_i$, and issues a verifiable credential $VC_i = \{DID_i, DID_{CA}, claim, Sig_{CA}(claim), Exp_i\}$ that vouches for $U_i$'s personal information, such as occupation, age, etc. Then, $CA$ sends $VC_i$ to $U_i$. After checking $VC_i$, $U_i$ computes $HVC_i = (VC_i) \oplus h(r_i||ID_i||HPW_i)$ and stores $\{DID_i, HVC_i, \delta_i\}$ in the device.

### 5.3. User Registration Phase

User $U_i$ registers with $SP$ using his/her own decentralized identifier. $SP$ verifies that the user's decentralized identifier is valid, and then the user's avatar is generated in virtual space. This phase is performed over a secure channel. Figure 3 shows the user registration phase and detailed processes steps are as follows.

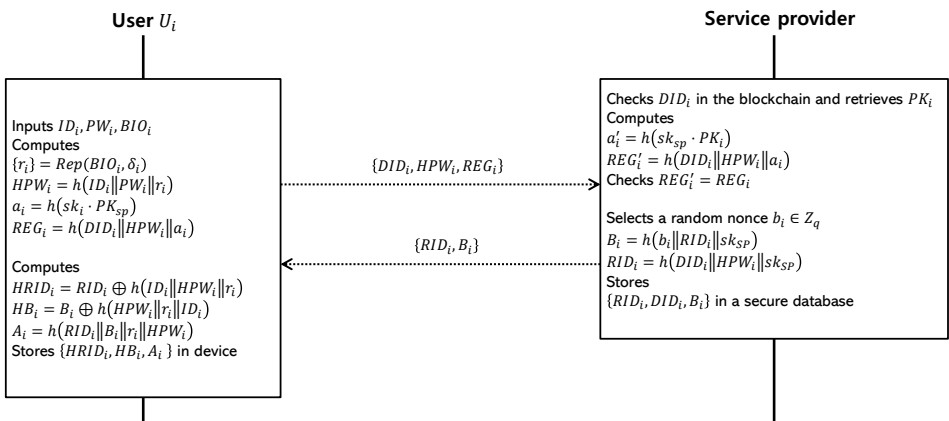

**Figure 3.** User registration phase of the proposed scheme.

- **UR-1:** $U_i$ inputs a identity $ID_i$, password $PW_i$, and imprints a biomatic information $BIO_i$. Then, $U_i$ computes $\{r_i\} = Rep(BIO_i, \delta_i)$, $HPW_i = h(ID_i||PW_i||r_i)$, $a_i = h(sk_i \cdot PK_{sp})$, $REG_i = h(DID_i||HPW_i||a_i)$, and send $\{DID_i, HPW_i, REG_i\}$ to $SP$.
- **UR-2:** $SP$ checks the validity of $DID_i$ and retrieves $PK_i$ from the blockchain. If it is valid, $SP$ computes $a_i = h(sk_{sp} \cdot PK_i)$, $REG_i' = h(DID_i||HPW_i||a_i)$ and verifies $REG_i \overset{?}{=} REG_i'$. If the equation is correct, $SP$ selects a random nonce $b_i \in Z_q$ and calculates $B_i = h(b_i||RID_i||sk_{sp})$, $RID_i = h(DID_i||HPW_i||sk_{sp})$. After that, $SP$ dispatches $\{RID_i, B_i\}$ to $U_i$ and stores $\{RID_i, DID_i, B_i\}$ in a secure database.
- **UR-3:** $U_i$ computes $HRID_i = RID_i \oplus h(ID_i||HPW_i||r_i)$, $HB_i = B_i \oplus h(HPW_i||r_i||ID_i)$, $A_i = h(RID_i||B_i||r_i||HPW_i)$ and stores $\{HRID_i, HB_i, A_i\}$ in $U_i$'s XR devices.

*5.4. Login Phase*

When the user $U_i$ attempts to access the $SP$, the user and $SP$ authenticate each other. If mutual authentication between the user and $SP$ is completed and the session key is established, the user and $SP$ communicate using the session key to guarantee secure communication. Figure 4 presents the login phase and the detailed processes of this phase are as follows.

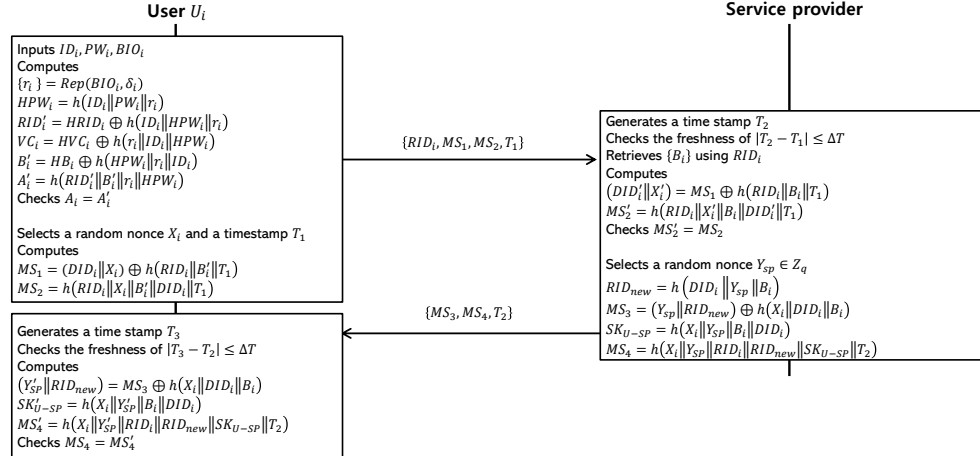

**Figure 4.** Login phase of the proposed scheme.

- **LA-1:** User $U_i$ first enters $ID_i$, $PW_i$, and $BIO_i$. Then, $U_i$ computes $\{r_i\} = Rep(BIO_i, \delta_i)$, $HPW_i = h(ID_i||PW_i||r_i)$, $RID_i' = HRID_i \oplus h(ID_i||HPW_i||r_i)$, $VC_i = HVC_i \oplus h(r_i||ID_i||HPW_i)$, $B_i' = HB_i \oplus h(HPW_i||r_i||ID_i)$, $A_i' = h(RID_i'||B_i'||r_i||HPW_i)$, and checks the $A_i = A_i'$. If the equation is correct, $U_i$ selects a random nonce $X_i$ and a current timestamp $T_1$, and computes $MS_1 = (DID_i||X_i) \oplus h(RID_i||B_i'||T_1)$, $MS_2 = h(RID_i||X_i||B_i'||DID_i||T_1)$. After that, $U_i$ sends $\{RID_i, MS_1, MS_2, T_1\}$ to $SP$.
- **LA-2:** $SP$ generates a current timestamp $T_2$ and checks the freshness of the timestamp. Next, $SP$ retrieves $\{B_i\}$ from the database using $RID_i$, and calculates $(DID_i'||X_i') = MS_1 \oplus h(RID_i||B_i||T_1)$, $MS_2' = h(RID_i||X_i'||B_i||DID_i'||T_1)$. $SP$ checks the $MS_2' \overset{?}{=} MS_2$, and selects a random nonce $Y_{sp} \in Z_q$ and calculates $RID_{new} = h(DID_i||Y_{sp}||B_i)$, $MS_3 = (Y_{SP}||RID_{new}) \oplus h(X_i||DID_i||B_i)$, $SK_{U-SP} = h(X_i||Y_{SP}||B_i||DID_i)$, $MS_4 = h(X_i||Y_{SP}||RID_i||RID_{new}||SK_{U-SP}||T_2)$. After that, $SP$ transmits $\{MS_3, MS_4, T_2\}$ to $U_i$.
- **LA-3:** After reception of the messages, $U_i$ checks the freshness of $T_2$ and computes $(Y_{sp}'||RID_{new}) = MS_3 \oplus h(X_i||DID_i||B_i)$, $SK_{U-SP} = h(X_i||Y_{sp}'||B_i||DID_i)$, $MS_4' = h(X_i||Y_{sp}'||RID_i||RID_{new}||SK_{U-SP}||T_2)$. Then, $U_i$ checks the validity of $MS_4 \overset{?}{=} MS_4'$, calculates $HRID_i' = RID_{new} \oplus h(ID_i||HPW_i||r_i)$, and updates $HRID_i$ with $HRID_i'$.

### 5.5. Avatar Authentication Phase

In the virtual space, user $U_i$ can interact with other avatars $U_j$. For secure avatar-to-avatar interactions, the user provides the verifiable credentials proving the personal information to perform the avatar authentication phase. Figure 5 shows the avatar authentication phase and the detailed steps are as follows.

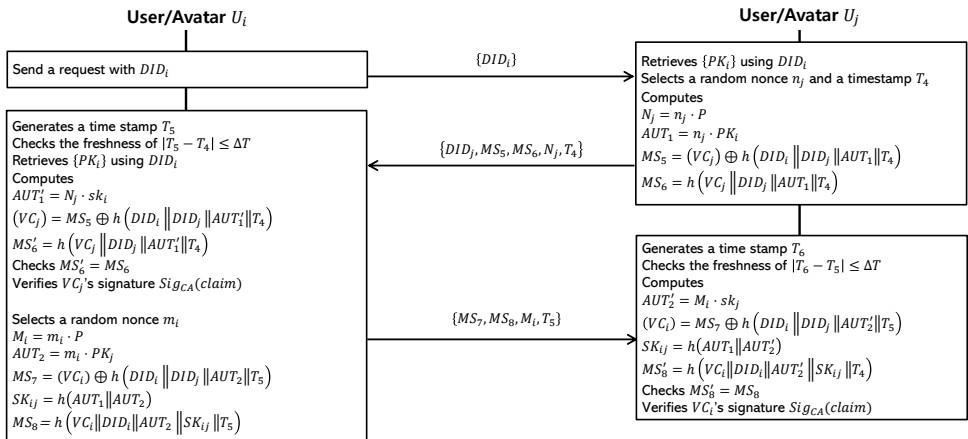

**Figure 5.** Avatar authentication phase of the proposed scheme.

- **AA-1:** $U_i$ first sends a request including $DID_i$ to $U_j$. After reception of the request, $U_j$ retrieves $\{PK_i\}$ using $DID_i$, and selects a random nonce $n_j$ and a current timestamp $T_4$. Next, $U_j$ computes $N_j = n_j \cdot P$, $AUT_1 = n_j \cdot PK_j$, $MS_5 = (VC_j) \cdot h(DID_i||DID_j||AUT_1||T_4)$, $MS_6 = h(VC_j||DID_j||AUT_1||T_4)$, and sends $\{DID_j, MS_5, MS_6, N_j, T_4\}$ to $U_i$.

- **AA-2:** After receiving the message $\{DID_j, MS_5, MS_6, N_j, T_4\}$, $U_i$ checks the validity of $T_4$, and retrieves $\{PK_j\}$ from the blockchain using $DID_j$. Then, $U_i$ computes $AUT_1' = N_j \cdot sk_i$, $(VC_j) = MS_5 \cdot h(DID_i||DID_j||AUT_1'||T_4)$, $MS_6' = h(VC_j||DID_j||AUT_1'||T_4)$ and verifies the equation $MS_6' \overset{?}{=} MS_6'$ and the signature $Sig_{CA}(claim)$ of the $VC_j$. Next, $U_i$ selects a random nonce $m_i$ and calculates $M_i = m_i \cdot P$, $AUT_2 = m_i \cdot PK_j$, $MS_7 = (VC_i) \cdot h(DID_i||DID_j||AUT_2||T_5)$, $MS_8 = h(VC_i||DID_i||AUT_2||h(AUT_1||AUT_2)||T_5)$. And $U_i$ transmits $\{MS_7, MS_8, M_i, T_5\}$ to $U_j$.

- **AA-3:** Upon reception of message $\{MS_7, MS_8, M_i, T_5\}$, $U_j$ checks the freshness of $T_5$ and computes $AUT_2' = M_i \cdot sk_j$, $(VC_i) = MS_7 \cdot h(DID_i||DID_j||AUT_2'||T_5)$, $MS_8' = h(VC_i||DID_i||AUT_2'||h(AUT_1||AUT_2')||T_4)$. Finally, $U_j$ checks that $MS_8' \overset{?}{=} MS_8$ is correct and verifies $VC_i$'s signature $Sig_{CA}(claim)$.

## 6. Security Analysis

In this section, we show the resilience of the proposed system against malicious security attacks through an informal analysis and AVISPA simulation. We also utilize BAN logic [30,31], which is a widely accepted formal security analysis, to prove that the proposed scheme is guaranteed for secure mutual authentication. Subsequently, we prove the session key secrecy utilizing the real-or-random (ROR) model.

### 6.1. Informal Security Analysis

We perform informal security analysis to demonstrate how the proposed protocol fulfills some of the security requirements, such as impersonation, replay, perfect forward secrecy, session key disclosure attacks, mutual authentication, etc.

#### 6.1.1. Stolen XR Device Attack

Under the assumptions in Section 4.1, an adversary $Adv$ can seize the user's XR device and extract the stored parameters $\{DID_i, HVC_i, \delta_i, HRID_i, HB_i, A_i\}$ to obtain sensitive information $VC_i, B_i$. However, all the stored sensitive information are masked with hash,

XOR operations utilizing identity $ID_i$, password $PW_i$, and biometric information $BIO_i$ so that the $Adv$ cannot obtain sensitive information. Thus, the proposed scheme is secure against stolen XR device attacks.

### 6.1.2. Offline Password-Guessing Attack

The $Adv$ attempts to guess the user's password $PW_i$ using extracted values from the $U_i$'s XR device and intercepts the transmitted messages on public channels. However, it is impracticable for $Adv$ to guess $PW_i$ without knowledge of the real identity $ID_i$ and response value $r_i$. $PW_i$ is constructed as $HPW_i = h(ID_i||PW_i||r_i)$, where $r_i$ is the response value from a fuzzy extractor with bio-information as the input. Therefore, our scheme is resistant to offline password-guessing attacks.

### 6.1.3. Impersonation Attack

$Adv$ can create fake login messages $\{RID_i, MS_1, MS_2, T_1\}$ and $\{MS_3, MS_4, T_2\}$ to impersonate legitimate user $U_i$ and gain unauthorized access to the metaverse environment supported by $SP$. However, $Adv$ cannot forge the request message and compute the session key $SK_{U-SP}$ because it is infeasible for $Adv$ to obtain $B_i$ and random nonces $X_i$ and $Y_{sp}$, where $B_i$, $X_i$, and $Y_{sp}$ are masked and $B_i$ is shared by $U_i$ and the $SP$ only. Therefore, the proposed protocol prevents impersonation attacks.

### 6.1.4. Avatar Impersonation Attack

In the metaverse, $Adv$ creates a fake avatar in an attempt to impersonate a legitimate user $U_i$'s avatar. $Adv$ should be required to prove ownership of the legitimate $U_i$'s decentralized identifier $DID_i$ and present verifiable credential $VC_i$ to others. However, $Adv$ cannot impersonate the legitimate user of the avatar because $Adv$ cannot obtain the private key corresponding to $DID_i$ and it is difficult to extract $VC_i$, which is masked with the real identity $ID_i$ and password $PW_i$. Furthermore, since the user can easily create a new DID, if a problem occurs with the existing DID, the user can obtain a new DID and VC and discard the existing DID. Therefore, the proposed scheme prevents an avatar impersonation attack.

### 6.1.5. Session Key Disclosure Attack

In the proposed scheme, $Adv$ should obtain the secret value $B_i$ and the random nonces $X_i$ and $Y_{sp}$ to compute a common session key. However, it is infeasible for $Adv$ to compute a valid session key $SK_{U-SP}$ because $U_i$'s secret value $B_i$ is masked with the real identity $ID_i$, password $PW_i$, and biomatic information $BIO_i$. In addition, random nonces $X_i$ and $Y_{sp}$ are masked with $B_i$ and $DID_i$. $Adv$ also cannot decrypt $M_1$ without $U_i$'s private key $r_{User}$. Therefore, the session key $SK_{U-SP} = h(X_i||Y_{SP}||B_i||DID_i)$ disclosure attacks are computationally infeasible in the proposed protocol.

### 6.1.6. Perfect Forward Secrecy

Even if the long-term secret keys $sk_i$ and $sk_{sp}$ are compromised, $Adv$ does not obtain the previous session key $SK_{U-SP} = h(X_i||Y_{SP}||B_i||DID_i)$. Since $DID_i$ and $B_i$ are not revealed in messages transmitted on public channels, and random nonces $X_i$ and $Y_{sp}$ are refreshed every session, $Adv$ cannot obtain the previous session key. Therefore, the proposed protocol guarantees perfect forward secrecy. Furthermore, if the secret key is compromised, the user can easily invalidate the existing DID associated with that key and create a new DID with a corresponding key pair. Subsequently, by re-registering with the system, the user can obtain a new VC from the $CA$.

### 6.1.7. Replay Attack and MITM Attack

$Adv$ attempts replay and man-in-the-middle (MITM) attacks using previously transmitted messages. However, all the transmitted messages include the current timestamps $T_x$ are refreshed with each session, and $U_i$ and $SP$ check the freshness of all transmitted messages. In addition, $RID_i$ is also updated every session. If the received messages are in-

valid, the receiver terminates the current session. Therefore, the proposed protocol prevents replay and MITM attacks.

### 6.1.8. Insider Attack

According to Section 4.1, an internal $Adv$ attempts to impersonate $U_i$'s avatar using a fake avatar and intercepted messages $DID_i$, $\{DID_j, MS_5, MS_6, N_j, T_4\}$ and $\{MS_7, MS_8, M_i, T_5\}$. However, it is infeasible for $Adv$ to calculate $AUT_1 = N_j \cdot PK_j = N_j \cdot sk_i$, $AUT_2 = m_i \cdot PK_j = M_i \cdot sk_j$ without the private keys $sk_i$, $sk_j$ and random nonces $n_j$ and $m_i$. Thus, $Adv$ cannot obtain verifiable credential $VC$ without $AUT_1$, $AUT_2$. Therefore, $Adv$ cannot disguise itself as another legitimate user in the metaverse without private key $sk_i$ and $VC_i$ corresponding to $DID_i$.

### 6.1.9. Ephemeral Secret Leakage Attack

According to Section 4.1, $Adv$ can obtain the ephemeral secret values, such as $X_i$ and $Y_{sp}$. Then, the adversary can attempt to calculate the session key $SK_{U-SP}$. However, $Adv$ cannot calculate $SK_{U-SP}$ without $B_i$ and $DID_i$. Therefore, the proposed protocol has resistance to the ephemeral key leakage attack.

### 6.1.10. Mutual Authentication

Sections 6.1.3 and 6.1.5 demonstrate that $Adv$ cannot impersonate $U_i$ and obtain the session key. In the login phase, $U_i$ and $SP$ verify all transmitted messages. When $SP$ receives the login request message $\{RID_i, MS_1, MS_2, T_1\}$ from $U_i$, $SP$ verifies $MS'_2 \stackrel{?}{=} MS_2$. If valid, $SP$ authenticates $U_i$. When $U_i$ receives response messages $\{MS_3, MS_4, T_2\}$ from $SP$, $U_i$ verifies the equation $MS'_4 \stackrel{?}{=} MS_4$. If valid, $U_i$ authenticates $SP$. Consequently, all entities are mutually authenticated so that the proposed system provides secure mutual authentication.

### 6.1.11. Anonymity

If $Adv$ intercepts, modifies, and deletes the transmitted messages, it can execute Section 6.1.1 to extract $U_i$'s real identity. However, it is impossible for $Adv$ to obtain real identity $ID_i$. The user's $ID_i$ is comprised of $RID_i = h(DID_i||HPW_i||sk_{sp})$ by using hash and XOR functions. Therefore, the proposed protocol ensures the anonymity of $U_i$.

### 6.1.12. Privacy-Preservation

In the proposed scheme, $U_i$'s identity and sensitive personal information are managed by the user, and it is provided only to other relevant parties when access to specific services and data is required. The $SP$ can only check some of $U_i$'s information as a requirement to access the metaverse environment, and $U_i$'s other information cannot be viewed without user consent. Therefore, the proposed scheme guarantees the privacy preservation of the user.

### 6.1.13. Untraceability

Nontraceability ensures that an external $Adv$ cannot track the legitimate user $U_i$. Because all messages are dynamic and unique using temporary identities $RID_x$, random nonces $X_i$ and $Y_j$, and timestamps $T_x$ in each session, where each parameters are updated every session in the login phase, the proposed scheme provides untraceability for $U_i$.

### 6.1.14. Denial-of-Service (DoS) Attack

The $Adv$ attempts to create a number of login request messages and transmit them to the $SP$ to paralyze the network. However, since the $SP$ checks the $RID_i$ and $T_i$, which are updated each session, the $Adv$ cannot create new valid messages. Even if the $Adv$ attempts to resend past messages, $SP$ considers them invalid and terminates the connection. Therefore, the proposed scheme ensures safety against DoS attacks.

### 6.2. Security Analysis Using BAN Logic

Over the BAN logic analysis, we prove that the proposed scheme guarantees secure mutual authentication between the user $U_i$ and $SP$. We also define the rules, goals, idealized forms, and assumptions for performing BAN logic analysis. Table 2 introduces the BAN logic notations.

**Table 2.** Notations for BAN logic.

| Notation | Description |
|---|---|
| $\alpha \mid \equiv X$ | $\alpha$ **believes** statement $X$ |
| $\#X$ | Statement $X$ is **fresh** |
| $\alpha \lhd X$ | $\alpha$ **sees** statement $X$ |
| $\alpha \Rightarrow X$ | $\alpha$ **controls** statement $X$ |
| $\alpha \mid \sim X$ | $\alpha$ once **said** $X$ |
| $\{X\}_K$ | $X$ is **encrypted** under key $K$ |
| $< X >_Y$ | Formula $X$ is **combined** with formula $Y$ |
| $\alpha \overset{K}{\leftrightarrow} \beta$ | $\alpha$ and $\beta$ may use **shared key** $K$ to communicate |
| $\overset{K}{\longrightarrow}\beta$ | $\beta$ has $K$ as a **public key** |
| $SK$ | Session key used in the current session |

BAN Logic Rules

The BAN logic rules are as follows:

1. Message meaning rule:

$$\frac{\alpha \mid \equiv \alpha \overset{K}{\leftrightarrow} \beta, \quad \alpha \lhd \{X\}_K}{\alpha \mid \equiv \beta \mid \sim X}$$

2. Nonce verification rule:

$$\frac{\alpha \mid \equiv \#(X), \quad \alpha \mid \equiv \beta \mid \sim X}{\alpha \mid \equiv \beta \mid \equiv X}$$

3. Jurisdiction rule:

$$\frac{\alpha \mid \equiv \beta \mid \Longrightarrow X, \quad \alpha \mid \equiv \beta \mid \equiv X}{\alpha \mid \equiv X}$$

4. Freshness rule:

$$\frac{\alpha \mid \equiv \#(X)}{\alpha \mid \equiv \#(X,Y)}$$

5. Belief rule:

$$\frac{\alpha \mid \equiv (X,Y)}{\alpha \mid \equiv X.}$$

### 6.3. Goals

We present the following security goals to show that the proposed system guarantees a secure mutual authentication.

**Goal 1:** $User \mid \equiv (User \overset{SK}{\longleftrightarrow} SP)$

**Goal 2:** $User \mid \equiv SP \mid \equiv (User \overset{SK}{\longleftrightarrow} SP)$

**Goal 3:** $SP \mid \equiv (User \overset{SK}{\longleftrightarrow} SP)$

**Goal 4:** $SP \mid \equiv User \mid \equiv (User \overset{SK}{\longleftrightarrow} SP)$

### 6.3.1. Idealized Forms

The idealized forms are the following:

$Msg_1$: $User \rightarrow SP$: $(RID_i, MS_1, MS_2, T_2)_{B_i}$
$Msg_2$: $SP \rightarrow User$: $(MS_3, MS_4, T_2)_{B_i}$

### 6.3.2. Assumptions

We define the following initial assumptions for the BAN logic proof.

$A_1$: $SP \mid\equiv \#(T_1)$

$A_2$: $User \mid\equiv \#(T_2)$

$A_3$: $User \mid\equiv (SP \xleftrightarrow{B_i} User)$

$A_4$: $SP \mid\equiv (User \xleftrightarrow{B_i} SP)$

$A_5$: $SP \mid\equiv \#(X_i)$

$A_6$: $User \mid\equiv \#(Y_{sp})$

$A_7$: $User \mid\equiv SP \Rightarrow (User \xleftrightarrow{SK} SP)$

$A_8$: $SP \mid\equiv User \Rightarrow (User \xleftrightarrow{SK} SP)$

### 6.3.3. Proof Using BAN Logic

The detailed steps of the BAN logic proof are as follows:

**Step 1:** From $Msg_1$,
$$S_1 : SP \triangleleft (RID_i, MS_1, MS_2, T_2)_{B_i}$$

**Step 2:** Upon the message meaning rule with $S_1$ and $A_4$,
$$S_2 : SP \mid\equiv User \sim (RID_i, MS_1, MS_2, T_2)$$

**Step 3:** Using the freshness rule with $A_1$,
$$S_3 : SP \mid\equiv \#(RID_i, MS_1, MS_2, T_2)$$

**Step 4:** Using the nonce verification rule with $S_2$ and $S_3$,
$$S_4 : SP \mid\equiv User \mid\equiv (RID_i, MS_1, MS_2, T_2)$$

**Step 5:** Since the session key $SK_{U-SP} = h(X_i||Y_{SP}||B_i||DID_i)$, from $S_4$ and $A_5$,
$$S_5 : SP \mid\equiv User \mid\equiv (User \xleftrightarrow{SK} SP) \textbf{ (Goal 4)}$$

**Step 6:** Upon the jurisdiction rule with $S_6$ and $A_8$,
$$S_6 : SP \mid\equiv (User \xleftrightarrow{SK} SP) \textbf{ (Goal 3)}$$

**Step 7:** Using the $Msg_2$,
$$S_7 : User \triangleleft (b_1, ID_{SP}, T_2)_{a_1}$$

**Step 8:** From the message meaning rule with $S_8$ and $A_3$,
$$S_8 : User \mid\equiv SP \sim (b_1, ID_{SP}, T_2)_{a_1}$$

**Step 9:** Using the freshness rule with $A_2$,

$$S_9 : User \models \#(b_1, ID_{SP}, T_2)_{a_1}$$

**Step 10:** Upon the nonce verification rule with $S_9$ and $S_{10}$,

$$S_{10} : User \models SP \models (b_1, ID_{SP}, T_2)_{a_1}$$

**Step 11:** Since the session key $SK_{U-SP} = h(X_i||Y_{SP}||B_i||DID_i)$, from $S_{11}$ and $A_6$,

$$S_{11} : User \models SP \models (User \xleftrightarrow{SK} SP) \textbf{ (Goal 2)}$$

**Step 12:** Utilizing the jurisdiction rule with $S_{13}$ and $A_7$,

$$S_{12} : User \models (User \xleftrightarrow{SK} SP) \textbf{ (Goal 1)}$$

Therefore, the proposed protocol achieves secure mutual authentication between the user and SP.

*6.4. ROR Model*

The ROR model, which is based on probabilistic game theory, is widely used to analyze the semantic security of an authenticated key agreement [32–34]. Using the ROR model, we demonstrate that our proposed scheme ensures session key security against a malicious adversary within probabilistic polynomial time. We first present the fundamentals of the ROR model in Table 3. We follow this by proving the session key security of our proposed scheme.

**Table 3.** Various queries and descriptions.

| Query | Description |
|---|---|
| $Execute(\mathcal{P}_U^t, \mathcal{P}_{SP}^t)$ | $\mathcal{A}$ using this query to tap the communication messages transmitted between $\mathcal{P}_U^t$ and $\mathcal{P}_{SP}^t$. |
| $Send(\mathcal{P}^t, M)$ | $\mathcal{A}$ sends a messages to the $\mathcal{P}^t$ and receives a response messages from $\mathcal{P}^t$. |
| $Reveal(\mathcal{P}^t)$ | $\mathcal{A}$ gets a current session key between $\mathcal{P}^t$ and its partner. |
| $Test(\mathcal{P}^t)$ | $\mathcal{A}$ guesses the probabilistic outcome for a flipped unbiased coin $C$. If the session key is fresh, $\mathcal{A}$ receives $C = 0$. If the session key is not fresh, $\mathcal{A}$ receives $C \neq 0$. Otherwise, $\mathcal{A}$ obtains null value ($\perp$). |
| $Corrupt(\mathcal{P}_U^t)$ | This query presumes an active attack. $\mathcal{A}$ extracts secret values stored in the XR devices by executing a power analysis. |

In the ROR model, adversary $\mathcal{A}$ interacts with the $t-$th instance of an executing participant, $\mathcal{P}^t$. Then, we define $\mathcal{P}_U^t$ and $\mathcal{P}_{SP}^t$ as the participants of $t$-th $U_i$ and $t$-th $SP$. In the ROR model, the adversary can execute *Execute*, *Send*, *Reveal*, *Test*, and *Corrupt* to consider different queries presuming actual security attacks. The descriptions of each query are introduced in Table 3. Furthermore, a query of the collision-resistant one-way hash function is denoted as *Hash*.

**Theorem 1.** *Before proving the session key security of the proposed scheme, we define $q_{hash}$ and $q_{send}$ as the number of Hash and Send queries, and $|Hash|$ as the range space of the hash function. $C$ and $s$ denote Zipf's parameters [35], and $l_B$ is the number of bits in the biometric secret key $r_i$. When adversary $\mathcal{A}$ obtains the session key in polynomial time, the adversary $\mathcal{A}$ breaches the*

*semantic security of the proposed scheme, and its advantage is represented by $Adv_{\mathcal{A}}(t)$. $Adv_{\mathcal{A}}(t)$ is estimated by*

$$Adv_{\mathcal{A}}(t) \leq \frac{q_{hash}^2}{|Hash|} + 2\,max\{C' \cdot q_{send}^s, \frac{q_{send}}{2^{l_B}}\}. \tag{1}$$

**Proof.** We consider the following games $G_i, i = [0, 3]$, and assume that $Pr[Succ_{G_i}]$ is $\mathcal{A}$'s advantage of winning the game $G_i$. The detailed descriptions of each game are discussed as follows. □

- **Game 0:** $G_0$ presents the $\mathcal{A}$'s real attacks against our proposed scheme in the ROR model. $\mathcal{A}$ selects the bit $c$ at the starting of $G_0$. $Adv_{\mathcal{A}}(t)$ is as follows.

$$Adv_{\mathcal{A}}(t) = |2Pr[Succ_{G_0}] - 1|. \tag{2}$$

- **Game 1:** $G_1$ is modeled such that $\mathcal{A}$ implements an eavesdropping attack. In this game, $\mathcal{A}$ executes the $Execute(\cdot)$ query to steal the communicated messages $\{RID_i, MS_1, MS_2, T_1\}$ and $\{MS_3, MS_4, T_2\}$ between $U_i$ and $SP$. At the end of this game, $\mathcal{A}$ executes $Reveal$ and $Test$ queries to check whether the derived session key $SK_{U-SP}$ is an actual or random key. $\mathcal{A}$ needs the long-term secret values (such as the private keys $sk_i$ and $sk_{sp}$), and the short-term secret values (such as the random nonces $X_i$ and $Y_{sp}$) to extract the $SK_{U-SP}$. However, it is impracticable for $\mathcal{A}$ to obtain these secret values, even if $\mathcal{A}$ obtains all communicated messages. As shown, the eavesdropping messages $\{RID_i, MS_1, MS_2, T_1\}$ and $\{MS_3, MS_4, T_2\}$ do not increase the probability of a winning game $G_1$. Therefore, because games $G_1$ and $G_0$ are indistinguishable, we obtain

$$Pr[Succ_{G_1}] = Pr[Succ_{G_0}]. \tag{3}$$

- **Game 2:** $G_2$ is modeled as an active attack. In this game, $\mathcal{A}$ executes the $Send$ and $Hash$ queries to guess the hash collision. However, all exchanged messages are protected using the one-way hash function $h(\cdot)$ and consist of secret credentials and random numbers. Moreover, it is difficult for $Adv$ to derive secret credentials and a random nonce because it is a computationally infeasible problem depending on the properties of $h(\cdot)$. So, using the birthday paradox, we obtain the following inequality:

$$|Pr[Succ_{G_1}] - Pr[Succ_{G_2}]| \leq \frac{q_{hash}^2}{2|Hash|}. \tag{4}$$

- **Game 3:** $G_3$ is modeled such that an active attack is implemented by $\mathcal{A}$. In this game, $\mathcal{A}$ executes the $Corrupt(\mathcal{P}_V^t, \mathcal{P}_{EP}^t)$ query to extract the secret values $\{DID_i, HVC_i, \delta_i, HRID_i, HB_i, A_i\}$ from the user's XR devices. Subsequently, to derive credential $VP_i$ and $U_i$'s secret key $sk_i$, $\mathcal{A}$ must guess the unknown password $PW_i$ through operating the $Send$ query. However, it is computationally infeasible for $\mathcal{A}$ to guess the password $PW_i$ through the $Send$ query without $V_i$'s identity $ID_i$ and secret nonce $x_i$. In the absence of password-guessing attacks, games $G_2$ and $G_3$ are identical. The probability of $\mathcal{A}$ winning the game $G_4$ using Zip's law is

$$[Pr[Succ_{G_3}] - Pr[Succ_{G_4}] \leq max\{C' \cdot q_{send}^s, \frac{q_{send}}{2^{l_B}}\}. \tag{5}$$

After all of the games are executed, $\mathcal{A}$ conjectures the correct bit $c$. Hence, we obtain

$$Pr[Succ_{G_3}] = \frac{1}{2}. \tag{6}$$

Considering Equations (2) and (3), we obtain

$$\frac{1}{2} Adv_{\mathcal{A}}(t) = |Pr[Succ_{G_0}] - \frac{1}{2}|$$
$$= |Pr[Succ_{G_1}] - \frac{1}{2}|. \tag{7}$$

Then, we consider Equations (4) and (5) and obtain the following inequality:

$$\frac{1}{2} Adv_{\mathcal{A}}(t) = |Pr[Succ_{G_1}] - Pr[Succ_{G_4}]|$$
$$\leq \frac{q_{hash}^2}{2|Hash|} + max\{C' \cdot q_{send}^s, \frac{q_{send}}{2^{l_B}}\}. \tag{8}$$

Consequently, the stipulated result $Adv_{\mathcal{A}}(t)$ is presented by multiplying both sides of Equation (8):

$$Adv_{\mathcal{A}}(t) \leq \frac{q_{hash}^2}{|Hash|} + 2 \, max\{C' \cdot q_{send}^s, \frac{q_{send}}{2^{l_B}}\}. \tag{9}$$

### 6.5. Avispa Simulation Tool

AVISPA is a well-known security simulation tool that analyzes the protocols' ability to resist replay and MITM attacks [36–38]. The AVISPA tool employs the high-level protocols specifications language (HLPSL) for outlining the actions of each participant. Afterword, the HLPSL code of the protocol is converted into the intermediate format (IF) through the HLPSL2IF translator. Then, IF data are input to implement AVISPA on one of four backends, such as "the CL-based attack searcher (CL-AtSe)", "the on-the-fly-model checker (OFMC)", "the tree Automata-based protocol analyzer (TA4SP)", and "the SAT-based model checker (SATMC)". When IF data are passed through the selected backend, the simulation result is output following the output format (OF). In this paper, we perform AVISPA simulations of the proposed scheme using OFMC and the CL-AtSe backend, which provide the XOR operation. In OF, if the SUMMARY segment indicates SAFE, it means that the analyzed scheme is resistant to replay and MITM attacks.

Figure 6 describe the user's role in HLPSL code form. The other parties (service provider and certificate authority) are also coded in a format similar to Figure 6. Figure 7 indicates the goals and environment of the proposed protocol and the role of the session. Figure 8 presents the AVISAP simulation result of the proposed protocol using CL-AtSe and OFMC. The results under the CL-AtSe and OFMC backends show that the proposed protocol is safe. Therefore, the proposed protocol can be resilient against man-in-the-middle and replay attacks.

```
%%%%%%% Role UA %%%%%
role usera(UA,SP,CA : agent, SKuanc,SKuans :symmetric_key, H,ADD,MUL: hash_func, SND, RCV : channel(dy))
played_by UA
def=
local State: nat,
        IDi,PWi,BIOi,DIDi,PKi,SKi,RRi,INFOi,HPWi,HVCi,AAi,REGi,HRIDi,HBii,Aii,Xi,T1,MS1,MS2,SKus:text,

VCi,P: text,
        RIDi,SKsp,PKsp,BBi,Bii,Ysp,RIDnew,MS3,MS4,SKsu,T2:text

const sp1,sp2,sp3,sp4,ua_sp_xi,sp_ua_ysp: protocol_id
init State:=0
transition
%%%%% Set up phase %%%%%%%
1. State=0 /\RCV(start)=|>
State':=1 /\SKi':=new()
        /\DIDi':=new() /\RRi':=new() /\PKi':=MUL(SKi.P)
        /\SND({DIDi'.INFOi}_SKuanc)
        /\secret({IDi.PWi.BIOi.SKi'.RRi'},sp1,{UA})
        /\secret({INFOi},sp2,{UA,CA})
2. State=1 /\RCV({VCi'}_SKuanc)=|>
State':=2 /\HPWi':=H(IDi.PWi.RRi') /\HVCi':=xor(VCi',H(RRi'.IDi.PWi))
%%%%% Registration phase %%%%%
/\AAi':=H(MUL(SKi'.PKsp')) /\REGi':=H(DIDi'.HPWi'.AAi')
/\SND({DIDi'.HPWi'.REGi'}_SKuans)
3. State=2 /\RCV({H(DIDi'.H(IDi.PWi.RRi').SKsp).H(BBi'.H(DIDi'.H(IDi.PWi.RRi').SKsp).SKsp)}_SKuans)=|>
State':=3 /\HRIDi':=xor(H(DIDi'.H(IDi.PWi.RRi').SKsp),H(IDi.HPWi'.AAi))
        /\HBii':=xor(H(BBi'.H(DIDi'.H(IDi.PWi.RRi').SKsp).SKsp),H(HPWi'.RRi'.IDi))
        /\Aii':=H(H(DIDi'.H(IDi.PWi.RRi').SKsp).H(BBi'.H(DIDi'.H(IDi.PWi.RRi').SKsp).SKsp).RRi'.HPWi')
%%%%% Login phase %%%%%%
/\Xi':=new() /\T1':=new()
/\MS1':=xor(Xi',H(H(DIDi'.H(IDi.PWi.RRi').SKsp).H(BBi'.H(DIDi'.H(IDi.PWi.RRi').SKsp).SKsp).T1'))
/\MS2':=H(H(DIDi'.H(IDi.PWi.RRi').SKsp).Xi'.H(BBi'.H(DIDi'.H(IDi.PWi.RRi').SKsp).SKsp).DIDi'.T1')
/\SND(H(DIDi'.H(IDi.PWi.RRi').SKsp),MS1',MS2',T1')
/\witness(UA,SP,ua_sp_xi,Xi')
4. State=3 /\RCV(xor({Ysp'.H(DIDi'.Ysp'.Bii')},H(Xi'.DIDi'.Bii')),H(Xi'.Ysp'.RIDi'.H(DIDi'.Ysp'.Bii').H(Xi'.Ysp'.Bii'.DIDi').T2'),
T2')=|>
State':=4
        /\SKus':=H(Xi'.Ysp'.H(DIDi'.H(IDi.PWi.RRi').SKsp).H(DIDi'.Ysp'.Bii').DIDi')
        /\request(UA,SP,sp_ua_ysp,Ysp')
end role
```

**Figure 6.** Role of user.

```
%%%%%%%%%   session  %%%%%%
role  session(UA,SP,CA : agent, SKuanc,SKspnc,SKuans :symmetric_key, H,ADD,MUL:
hash_func)

def=
local SND1, SND2, SND3, RVC1, RVC2, RVC3 : channel(dy)
composition
usera(UA,SP,CA,SKuanc,SKuans,H,ADD,MUL,SND1,RCV1)
/\serviceprovider(UA,SP,CA,SKspnc,SKuans,H,ADD,MUL,SND2,RCV2)
/\certificateauth(UA,SP,CA,SKuanc,SKspnc,H,ADD,MUL,SND3,RCV3)
end  role

%%%%%  environments and goals  %%%%%%
role  environment()

def=
const ua,sp,ca : agent,
skuanc,skspnc,skuans :symmetric_key,
h,add,mul: hash_func,
idi,pwi,bioi,didi,pki,ski,rri,infoi,hpwi,hvci,aai,regi,hridi,hbii,aii,xi,t1,ms1,ms2,skus:text,
vci,p: text
ridi,skksp,pksp,bbi,bii,ysp,ridnew,ms3,ms4,sksu,t2:text
ua_sp_xi,sp_ua_ysp: protocol_id,
sp1,sp2,sp3,sp4: protocol_id

intruder_knowledge = {ua,sp,ca,didi,pki,pksp,ms1,ms2,ms3,ms4,t1,t2,h,add,mul}

composition
session(ua,sp,ca,skuanc,skspnc,skuans,h,add,mul)
/\session(i,sp,ca,skuanc,skspnc,skuans,h,add,mul)
/\session(ua,i,ca,skuanc,skspnc,skuans,h,add,mul)
/\session(ua,sp,i,skuanc,skspnc,skuans,h,add,mul)
end  role

goal
secrecy_of sp1,sp2,sp3,sp4
authentication_on ua_sp_xi
authentication_on sp_ua_ysp
end goal

environment()
```

**Figure 7.** Role of session, environment, and goal.

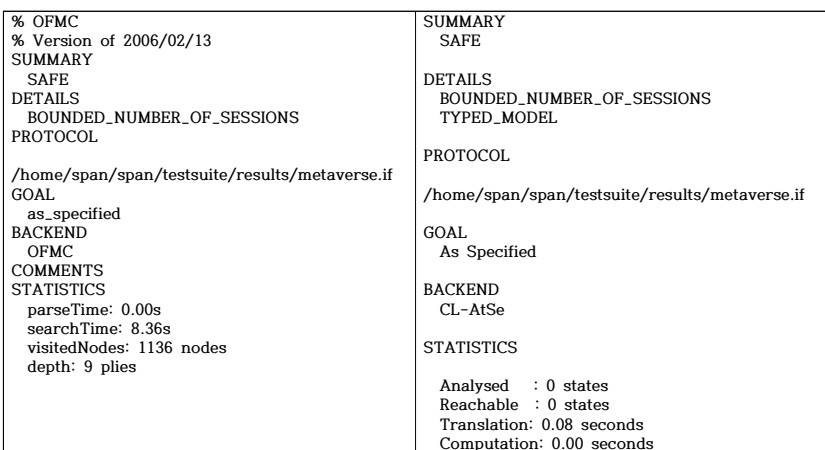

**Figure 8.** Result of AVISPA simulation.

## 7. Performance Analysis

We analyze the detailed comparative analysis of the proposed scheme with related schemes [16–18] in terms of the computation costs and the communication costs.

### 7.1. Analysis of Computation Cost

We compare the computation costs of the proposed scheme with the related schemes [16–18]. In this paper, we follow the execution time of the cryptographic operation measured by [39] using 2048 MB of RAM, Intel Pentium Dual CPU E2200 2.20 GHz, and the Ubuntu 12.04.1 LTS 32bit operating system. The cyclic group $G_1$ is a subgroup of $E(F_q) : y^2 = x^3 + x$, and $G_2$ is a subgroup of $F_q^2$. The group order of G1 is 160 bits, and the order of the base field is 512 bits. Depending on [39–41], we assume that the computation costs of 'a one-way hash function', 'biohasing function', 'elliptic curve point addition', 'elliptic curve scalar point multiplication', 'bilinear pairing', 'random nonce generation', and 'fuzzy extraction' are $T_H \approx 0.0023$ ms, $T_{BH} \approx 0.01$ ms [40], $T_{EA} \approx 0.0288$ ms, $T_{EM} \approx 2.226$ ms, $T_P \approx 5.811$ ms, $T_R \approx 0.539$ ms, $T_F \approx 2.68$ ms [41], respectively. We estimate the computation costs of the proposed scheme and related schemes and compare them. The comparison results are shown in Table 4. Because the proposed technique is designed based on XOR and Hash while minimizing the use of ECC, it shows much lower computation costs than the other existing schemes.

**Table 4.** Computation costs for authentication scheme: a comparative summary.

| Schemes | User | Service Provider |
| --- | --- | --- |
| Panda and Chattopadhyay [16] | $5T_{EM} + T_{EA} + 6T_H \approx 36.7759$ ms | $5T_{EM} + 2T_{EA} + 3T_H \approx 36.7837$ ms |
| Li et al. [17] | $7T_{EM} + 5T_H \approx 51.4723$ ms | $2T_P + 6T_{EM} + T_{EA} + 5T_H \approx 88.2458$ ms |
| Ryu et al. [18] | $4T_{EM} + T_{EA} + 8T_H + 2T_{BH} \approx 29.4438$ ms | $5T_{EM} + T_{EA} + 5T_H \approx 36.7755$ ms |
| The proposed scheme | $T_R + T_F + 11T_H \approx 3.2443$ ms | $T_R + 6T_H \approx 0.5528$ ms |

### 7.2. Analysis of Communication Cost

We assume that the bit sizes of the identity, hash output, random nonce, timestamp, and elliptic curve point are 160, 160, 160, 32, and 320, respectively. We present the comparison of the proposed scheme and existing schemes in Table 5. Under the results of the communication cost comparison, the proposed scheme provides a more efficient computation cost compared with the other existing schemes.

**Table 5.** Communication costs for each scheme: a comparative summary.

| Schemes | Costs |
|---|---|
| Panda and Chattopadhyay [16] | 1440 bits |
| Li et al. [17] | 1888 bits |
| Ryu et al. [18] | 1344 bits |
| Our scheme | 1024 bits |

### 7.3. Security and Functionality Comparison

In terms of security and functionality features, we compare the proposed scheme with other related schemes [16–18]. The security features of the proposed scheme and related schemes are presented in Table 6.

**Table 6.** A comparison of security and functionality features.

| | Panda and Chattopadhyay [16] | Li et al. [17] | Ryu et al. [18] | Our Scheme |
|---|---|---|---|---|
| Stolen IoT devices(XR) attack | − | − | √ | √ |
| Offline password guessing attack | √ | − | √ | √ |
| Impersonation attack | √ | √ | √ | √ |
| Avatar impersonation attack | − | − | √ | √ |
| Session key disclosure attack | √ | √ | √ | √ |
| Perfect forward secrecy | √ | √ | √ | √ |
| Replay attack | √ | √ | √ | √ |
| MITM attack | √ | √ | √ | √ |
| Insider attack | √ | − | √ | √ |
| Ephemeral secret leakage attack | × | √ | √ | √ |
| Mutual authentication | × | √ | √ | √ |
| Anonymity | √ | × | √ | √ |
| Privacy-preservation | − | − | × | √ |
| Untraceability | √ | × | √ | √ |
| Denial-of-Service (DoS) Attack | × | √ | × | √ |

√: scheme is secure or provides functionality feature ; ×: scheme is insecure and does not provide functionality feature; −: cannot be considered.

The results of our performance and security feature comparisons with related works indicate that our proposed scheme is more efficient in terms of computation and communication costs and satisfies a higher number of security requirements compared to existing schemes. Therefore, the proposed protocol can provide users with a secure service in the metaverse environment and is a lightweight protocol that takes into account the resource constraints of XR devices.

## 8. Conclusions

In this paper, we propose a secure authentication scheme for metaverse environments to provide a secure avatar interactions and prevent against various security attacks. In our scheme, users can utilize DID and VC to prove their identity to other avatars in the metaverse without revealing irrelevant personal information to service providers. Furthermore, the proposed scheme provides a secure communication channel against various attacks through secure authentication and key agreement between the user and service provider. The proposed scheme is resistant to various security attacks (including stolen XR devices, offline password guessing, user and avatar impersonation, etc.) by performing the ROR oracle security analyses, the well-known AVISPA simulation, and BAN logic analyses. Next, the proposed scheme provides lower computation and communication costs than other related schemes for the metaverse environment by the comparison of computation costs and communication costs. Therefore, the proposed scheme can be applied to practical metaverse environments to provide high security and privacy preservation. In the future, we intend to research authentication protocols for a secure and trusted metaverse environment, taking into consideration potential security issues that may arise in the blockchain.

**Author Contributions:** Conceptualization, M.K.; formal analysis, M.K. and S.S.; methodology, M.K. and Y.P. (Yohan Park); software M.K. and J.O.; validation, M.K., Y.P. (Yohan Park) and Y.P. (Youngho Park); writing—original draft, M.K.; writing—review and editing, J.K. and Y.P. (Youngho Park); supervision, Y.P. (Youngho Park). All authors have read and agreed to the published version of the manuscript.

**Funding:** This research was supported by the National Research Foundation of Korea (NRF) funded by the Ministry of Education under grant 2020R1I1A3058605.

**Data Availability Statement:** Not applicable.

**Conflicts of Interest:** The authors declare no conflict of interest.

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
