# Peer review of "Secure and Privacy-Preserving Authentication Scheme Using Decentralized Identifier in Metaverse Environment"

_electronics, doi:10.3390/electronics12194073_

Round 1

Reviewer 1 Report

The manuscript “Secure and privacy-preserving authentication scheme using decentralized identifier in metaverse environment” submitted for publication in Electronics presents an intriguing approach to addressing security and privacy concerns in the metaverse. Enhancing clarity, technical detail, and practical application discussions would strengthen the paper and its potential impact.

Blockchain, metaverses and authentication are three of the most significant research topics of modern society. The authors' work seeks to address the pressing security and privacy challenges in the metaverse, providing a novel authentication scheme that combines blockchain, decentralized identifiers, and verifiable credentials to enhance security while preserving user privacy. This work can have a meaningful impact on the evolving landscape of virtual reality and metaverse technologies. It seems very interesting and promising for future application. However, there are some elements that are still unclear and the manuscript lacks some research contents. The manuscript should be modified and improved with some revisions. In addition, my suggestions are as follows (see list below).

1. Clarity and Structure: The manuscript provides an informative overview of the metaverse, its potential applications, and the need for enhanced security and privacy. However, the structure could be improved for better readability. Consider restructuring the Introduction and Related Work sections to clearly outline the problem, existing solutions, and the specific contributions of your proposed scheme.

2. Motivation and Problem Statement: While you discuss the importance of security and privacy in the metaverse, it would be beneficial to provide specific use cases or examples that highlight the severity of the issues you aim to address. This can help readers better understand the real-world implications of your research.

3. Technical Details: The manuscript mentions the use of blockchain, decentralized identifiers (DID), and verifiable credentials (VC) but lacks in-depth technical explanations or references to relevant literature. Provide more technical details about how these technologies are integrated into your authentication scheme and refer to prior work or standards where applicable.

4. With the development of quantum computing, the classical cryptography system based on public and private keys has been threatened. Quantum cryptography provides a powerful security tool for privacy protection, blockchain, artificial intelligence and big data projects. The background of this article will be better organized if the authors add some important advances in the field of quantum digital signatures [Natl. Sci. Rev. 10, nwac228 (2023); npj Quantum Inf. 7, 98 (2021)], and quantum communication [Nature 557, 400 (2018); PRX Quantum 3, 020315 (2022)].

5. Conclusion and Practical Applications: The conclusion briefly summarizes the proposed scheme's benefits but could be expanded to discuss potential practical applications and future research directions. How can this scheme be practically implemented in metaverse environments, and are there any limitations or challenges to consider?

6. There are some spelling errors and grammatical errors in the article, please check and correct them. For example, “manipulator” in line 39, “key agreemet” in line 59, “we minimizes” in line 60 and so on.

Once the Authors have addressed the points that I have raised above I do not see any other reasons preventing this work from being published in Electronics.

Moderate editing of English language required

Reviewer 2 Report

The authors proposed a secure authentication scheme for metaverse environments to provide a secure avatar interactions and prevent against various security attacks. The  proposed scheme provides a secure communication channel against various attacks through secure authentication and key agreement between the user and the service provider. 

a. Consider expanding the discussion section to provide more in-depth analysis and interpretation of the results in relation to previous research or existing theories.

b. The conclusions part must be improved with the aid of drawing comparisons with similar works across the literature.

Overall all a nice and novel work done by authors.

Minor editing of English language required.

Reviewer 3 Report

The abstract should emphasise more on the challenges for readers to understand the necessity of the research. 

The Introduction may start by introducing the contexts related to the research, such as blockchain. Also, at the end of the section, the authors should define what the rest of the sections have to offer for readers to get an overview of the research work at this stage.

The related work section is insufficient and should be improved. I recommend referencing more recent works related to blockchain technology, security, and so on. The authors can consider the following papers on blockchain security;

- TRUSTEE: Towards the Creation of a Secure, Trustworthy, and Privacy-Preserving Framework.

- Assessing Blockchain Consensus and Security Mechanisms Against 51% Attacks.

Once again, Section 3 should include more context and must be improved. 

All the Figures/Tables should be defined in the text. Please review all figures/tables and correct where necessary.

Moreover, the Figures/Tables should appear right after they are referenced in the text and should be placed in the section where they are discussed.

For example, Figures 6, 7, 8 etc.

Please review all figures and place them appropriately. 

The Conclusion is satisfactory, however, the authors need to use the appropriate tense.

There are minor grammatical errors throughout the paper. Please review and correct where necessary. A few examples; 

a sensitive personal information→ sensitive personal information

managed by service provider → managed by the service provider

Round 2

Reviewer 1 Report

This manuscript can be accepted for publication.